# The length of a scroll: Quantitative evaluation of material reconstructions

Eshbal Ratzon[1]*, Nachum Dershowitz[2]

**1** The Department of Land of Israel Studies and Archaeology, Ariel University, Kiryat Hamada, Ariel, Israel,
**2** School of Computer Science, Tel Aviv University, Ramat Aviv, Israel

* eshbal@gmail.com

**Data Availability Statement:** All the relevant data are in the tables included in the paper.

**Funding:** ND recieved the awards: Grant BE 5916/1-1 KR 1473/8-1 from the Deutsch-Israelische Projektkooperation (DIP): https://www.dfg.de/foerderung/programme/inter_

## Abstract

Scholars have used mathematical models to estimate the missing length of deteriorated scrolls from ancient Egypt, Qumran, Herculaneum, and elsewhere. Based on such estimations, the content of ancient literature as well as the process of its composition is deduced. Though theoretically reasonable, many practical problems interfere with the method. In the current study, the empirical validity of these mathematical models is examined, showing that highly significant errors are quite frequent. When applied to comparatively intact scrolls, the largest contribution to errors is the subjectivity inherent in measuring patterns of damaged areas. In less well preserved scrolls, deterioration and deformation are more central causes of errors. Another factor is the quality of imaging. Hence, even after maximal reduction of interfering factors, one should only use these estimation methods in conjunction with other supporting considerations. Accordingly, past uses of this approach should be reevaluated, which may have substantial implications for the study of antiquity.

## Introduction

The length of a scroll . . . should not exceed the circumference,
>    nor should the circumference exceed the length.
>    What is a suitable length?
>    —Maimonides, *Mishneh Torah* (1180)

Many documents from the ancient world have been preserved in the form of a scroll, the most famous collections being the Dead Sea Scrolls and the Herculaneum papyri. The importance of such ancient scrolls for the study of the past cannot be overstated. Over time most of these important artifacts have suffered significant damage and require reconstruction. Thus, throughout the various fields of Egyptology, classics, and Judaic studies, scholars have endeavored to estimate the missing length of deteriorated scrolls.

Papyrus was invented in Egypt during the fourth millennium BCE. Numerous papyrus and some leather scrolls were found in Egypt originating from that time until the early centuries CE. These writings are significant witnesses for the history, literature, and religion of several cultures from the very beginning of antiquity.

foerdermassnahmen/deutsch_israelische_
projektkooperation/index.html and Grant \#1330/14
of the Israel Science Foundation (ISF): https://
www.isf.org.il/#/. The funders had no role in study
design, data collection and analysis, decision to
publish, or preparation of the manuscript.

**Competing interests:** The authors have declared
that no competing interests exist.

The Dead Sea Scrolls, dated to the turn of the Common Era, are of enormous historical significance, and their study continues to revolutionize our understanding of the evolution of Judaism and the emergence of Christianity. They are the oldest witnesses of the Biblical books and contain a treasure-trove of texts that have shed light on ancient Judaism shortly before and up into the time of Jesus and Paul. Unfortunately, the scrolls, or rather, the scroll fragments, were discovered in the prior century in very poor condition, having deteriorated over the millennia.

Almost two thousand Herculaneum papyri have been recovered, beginning in 1752, from a villa destroyed in the eruption of the Vesuvius volcano in the first century. This large library consisted of many classical works of literature and philosophy, including ancient scientific works. Some of these compositions were rediscovered in this collection after centuries of having been lost. Their additions to our knowledge of the classical world shed important light on the cradle of Western culture. But the papyri were crushed, carbonized, and decayed. Hence, ever since their discovery, various attempts have been made to unroll them and recover their textual contents.

This paper focuses on a well-established and popular mathematical method for reconstructing the length of a scroll based on its remaining fragments, a method already mentioned by the great Egyptologist, Ludwig Borchardt, in [1], and later developed and widely used in the disciplines of classics and religious studies for the reconstruction of disintegrated papyri and parchment scrolls found in various locations. Thus, the considerations discussed here are applicable to such reconstructions of any fragmentary ancient scroll.

Imagine a roll of wallpaper in a dry goods shop. After multiple purchases, the storekeeper wants to know how much material is left on the roll, though he has not kept track of prior purchases. It would be a major hassle to unroll all of it and measure. The preferable alternative is to measure the diameter or circumference of the remainder; then, knowing the thickness of the material and the size of the core, one can use a mathematical formula to reliably gauge the length of the remaining paper on the roll. (For one example out of many, see https://handymath.com/cgi-bin/rollen.cgi?submit=Entry by Roseller Sunga.) Textual scholars have tried to do something similar with fragmentary ancient scrolls, estimating the outer circumference and thickness of the substrate, and then calculating the length of the missing remainder. Unfortunately, the uncertainties and assumptions involved in the measuring process render the resulting estimates fairly unreliable.

Only very few of the Dead Sea Scrolls found in Qumran and its environs have been preserved relatively intact. The vast majority of those manuscripts have unfortunately deteriorated over the millennia and survive only as scattered fragments. From these fragments, scholars attempt to reconstruct as much as possible of each original scroll. They also often try to estimate its original overall length, including missing columns, so as to garner insight into the scope of the original work. In many cases, it is obvious that degradation began while the scroll was still rolled up, thus creating repeating patterns of damage around the same physical loci. Based on this observation, Hartmut Stegemann standardized a "geometrical" method to help reconstruct scrolls from their disconnected fragments. He first applied this method to *Hodayot* (in his unpublished 1963 dissertation, followed by [2, 3]); a detailed description of the method was published in [4].

This methodology has been further developed and has been used by others to estimate the circumference of a single turn of a scroll [4–14] thus providing an approximation of the length of the remaining inward part of the spiral, usually with the goal of estimating the amount of lost text [3, 11, 14–30]. Some data of this nature have been collected by Emanuel Tov [31] and have been used on occasion to interpret the social or religious role of particular scrolls [31–33].

According to Stegemann [4], if a comparatively large fragment with three consecutive points of damage is preserved, it is possible to learn the direction in which the scroll was rolled (normally—with the beginning of the composition to the outside—or occasionally the other way—with the beginning inward), since outer turns have greater circumference than inner ones. The measurements of the distances between corresponding points within these damages give the circumference of the scroll at that point.

The difference between those distances reveals how tightly the scroll was rolled. One then computes the length of the scroll from the measured fragment inward to the end of the text. It should be stressed that the method does not presume to estimate the overall original length of the entire scroll, but only its length from the outermost edge of the preserved fragment inwards to the center of the scroll. There is no way to measure missing outer parts, if any. The details of the calculations are explicated in Section Circle Approximation.

To facilitate the mathematical computations, Stegemann and scholars following in his footsteps have used an approximated model of concentric cylinders, thus assuming that the radius and circumference of consecutive rolls grow linearly, at a constant rate. Based on Stegemann's method, Dirk Stoll [6] offered an equivalent step-by-step procedure to derive the length of the original scroll from the circumference of its outermost turn; see Section Circle Approximation for details. This concentric-circle method is compared with a more precise spiral calculation, also suggested in the literature, in the Supporting information, Section S1 Appendix: Spiral Approximation in S1 File. The differences between the two are negligible.

Similar methods are employed by papyrologists for reconstructing missing sections of dismembered scrolls, such as those found in Herculaneum and Egypt—but with several significant differences. While most of the Dead Sea scrolls were found in multiple fragments, Herculaneum papyri were usually recovered still rolled, but damaged, and often with several fragments detached from the outermost part of the scroll. Thus, whereas Qumran scholars compute the remainder of the scroll inwards based on three or more damage points, Herculaneum scholars have the inner part of a scroll, and try to estimate the missing section between it and an extant but separated outer fragment. Accordingly, papyrologists may have more information available to estimate the length of a shorter missing section. Essler [34] formulates a slightly different mathematical method than Stegemann's for reconstructing the Herculaneum papyri; others developed similar methods for additional papyrological fields. Essler relies on similar formulas to those used in [35, p.149] to reconstruct the circumference of an Oxyrhynchus scroll with a given length. Hoffmann [36] and Janko [37, p.108] use a method closer to that used by Stegemann and Stoll for reconstructing an Egyptian Demotic and a Herculaneum papyrus, respectively. In Section S2 Appendix: Wedge Approximation in S1 File, this method is compared with Stegemann's by adapting the former to the Qumran situation, assuming three consecutive damages so as to reconstruct the rest of the scroll from that point to its center. Obviously, scholars make use of other, non-geometrical reconstruction methods in conjunction (or not) with the "Stegemann method," such as estimating the remaining length based on other known copies of the text—when available. But our concern here is only with the accuracy of estimates when based solely on physical and geometrical considerations. Combining methodologies is likely to lead to more accurate results than any one method alone.

## Length approximation

Already Stegemann [4] in his influential article referred to variations between different scrolls in thickness of the skins and tightness of their rolling. He suggests using the information that he collected regarding the average growth of circumference for different scrolls to give a range for the estimation of the minimum and maximum length of a reconstructed scroll. By his

calculation, this consideration added up to a 20–30% margin of error. Most scholars following him ignored this potential error in their work, simply giving the average reconstructed length for each scroll. Hoffmann [36] provided some considerations for estimating the potential error; he also refers to the problem of inconsistent growth of circumference, but reckons it does not contribute much to the overall error. Longacre [14] is exceptional in his more critical approach to the method in its entirety. His estimate of the the method's margin of error is based on theoretical considerations. Choosing different values for the outer and inner radius and for its growth in every layer, he compares the minimal and maximal lengths and arrives at ± 50% error.

In what follows, we continue along these lines, but offer an estimation of the minimal margin of error based on empirical measurements, rather than purely theoretical calculations. We first examine the question to what extent theory jibes with reality (in the next section) and then—based on images of long scrolls that are available today, described in Section Available Images—go on to approximate the margin of error in Results. We empirically examine these reconstruction methods from two perspectives:

1. While theoretically the methodology is sound, in reality repeating patterns of damage cannot always reliably indicate the circumference of the scroll at that point.

2. During the years since unearthing, scrolls presumably continue deteriorating, shrinking, and suffering from various other sorts of damage. Unfortunately, scholars often have nothing better than the available images—mostly only recently acquired—to rely on for measurements.

For most of the Dead Sea scrolls held by the Israel Antiquities Authority (IAA), two sets of images are available: old images taken at the Palestine Archaeological Museum (PAM) between 1947 and 1969 and new (IAA) images taken with state-of-the-art technology during the last decade for the Leon Levy Dead Sea Scrolls Digital Library (LLDSSDL). There are also additional scrolls held by other institutions that were imaged differently, the most important of which are the large scrolls safeguarded at the Shrine of the Book in the Israel Museum in Jerusalem. While all sets of images were very cautiously taken by talented experts of their times, for our purposes, each suffers from some disadvantages, as will be explained in Section Available Images.

We have measured the distances between consecutive damage points in three scrolls that remain comparatively intact: *Community Rule* or *Serekh haYahad* [= 1QS]; the *Great Psalms Scroll* [= 11QPs = 11Q5]; and *Apocryphal Psalms* [= 11QApocPs = 11Q11]. (The initial number in the scroll codes indicates the number of the cave in which the scroll was found, the letter Q stands for "Qumran," and the final element is either a serial number of the scroll or an abbreviation for the name of the work it contains. Regarding each segment as if it were the only surviving piece of the scroll, we have computed the length of the scroll in accordance with the standard method. The best results were found for 11Q5 with approximately 40% error and up to 250% in the worst-case scenario; the poorest results were found for 1QS with an average of around 240% error and deviation from the actual length of up to 1800%! For 11Q11, several measurements suggested that the scroll had been stored rolled with its beginning on the inside, a result that renders the entire calculation meaningless. Consequently, we maintain that no valid conclusion about the original scroll length can be drawn from the application of the technique as it currently stands. We suggest several means to reduce the margin of error, but it remains very high nevertheless.

It should be borne in mind that Stegemann's method is intended mainly to deal with a different problem altogether: the reconstruction of the order and placement of scattered fragments. This paper does not deal with that aspect of the theory.

### Theory vs. reality

Stegemann's reconstruction theory for scrolls is predicated on several crucial assumptions:

1. The repeating patterns of damage that are currently visible were introduced while the scroll was still rolled.

2. The distance between consecutive damage points is the actual circumference of the scroll at that point.

3. The increases in diameter of consecutive turns of the scroll were constant at the time the damage was incurred.

Under these assumptions, the length of the scroll from the surviving fragment inwards to its core can be estimated based on the distances between three damage points of a surviving fragment.

## Materials and methods

### Available images

For the sake of their conservation, the original objects of the DSS (like many other ancient artifacts) are no longer physically accessible to scholars, hence our entire experiment was based on the available high-quality images of the scrolls. Thus, Assumption C, that the diameter of turns of a scroll increases evenly, can only be evaluated after we look into the problems of the images themselves and the margin of error created by using them. As already mentioned, two large sets of images are available: the older infrared PAM images and newer multispectral IAA images, plus a smaller, but quite important, one from the Shrine of the Book.

**Palestine Archaeological Museum images.** The PAM images were taken by photographer, Najib Anton Albina, at the Palestine Archaeological Museum (colloquially the "Rockefeller Museum," and now formally so) between 1947 and 1967 (a few last photographs were made in 1969 by another photographer upon John Strugnell's request see [38, p.128]). Their big advantage is that they document the scrolls at a very early stage after their discovery, that is, before suffering further damage out of their original dry cave environment. While Albina was a highly competent professional photographer, specializing in the imaging of the Qumran scrolls, many contemporary advanced technologies were not yet available in his time. In particular, the PAM images' resolution is not nearly as good as can be achieved today. See Fig 1B and 1A for examples. In addition, the scaling of the images is a serious issue. While a ruler was placed on nearly every plate, it was hand drawn. The ruler itself was replaced many times over the twenty years of imaging, and was not always placed at the same location on the plate [38, p.125]. Furthermore, many fragments belonging to the same scroll are spread out over several different plates. So, to reconstruct the scroll from all its fragments, a scholar should rescale each plate, cut out the needed fragments from its photograph, and then place the cutouts on the same canvas. Alas, this process is prone to large errors due to the scaling differences.

**Israel Antiquities Authority images.** In 2012, the Leon Levy Dead Sea Scrolls Digital Library, led by the IAA scrolls' curator Pnina Shor, embarked on an ambitious project to image each and every fragment of the Dead Sea Scrolls using advanced multispectral imaging technology, a project that is nearing its successful conclusion [39]. The recto and verso of each fragment is imaged in multiple (visible and infrared) wavelengths and from various illuminating angles. This process results in a battery of digital images for both sides of each fragment, providing scholars with a robust and reliable inventory of graphic data. Out of this large resource of images, the multispectral images, near-infrared images, and raking-light images are most important for the task of material reconstruction. The new images solve most, if not

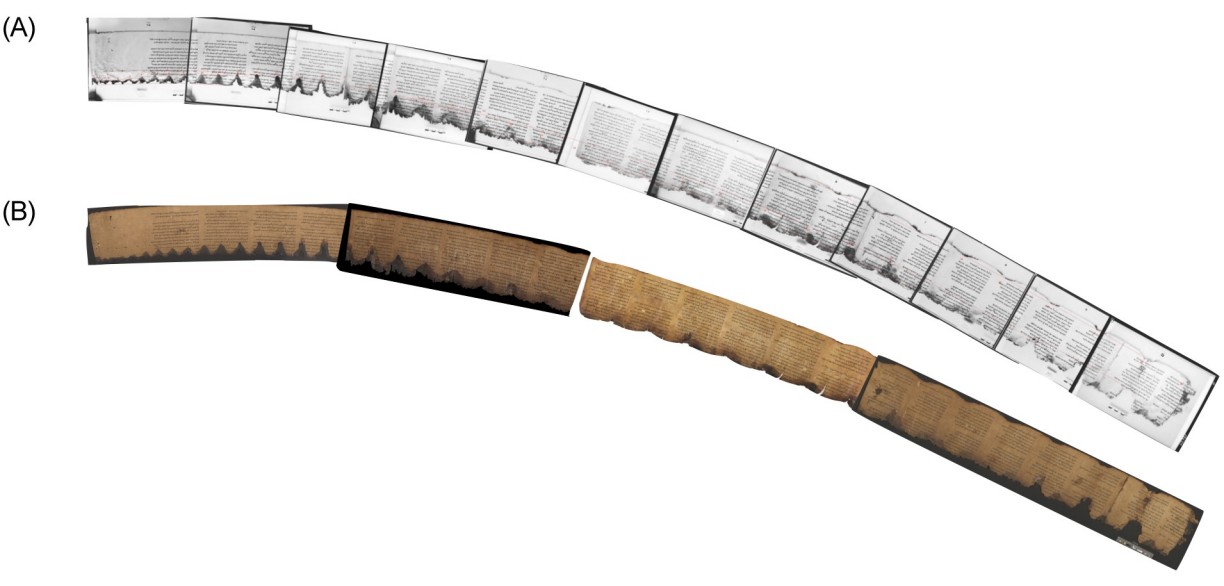

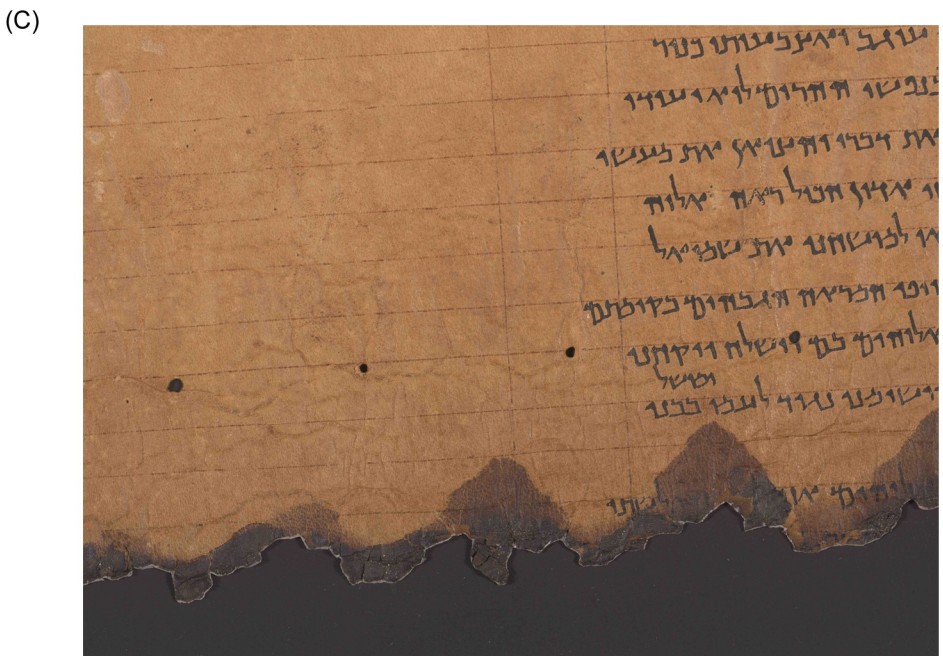

**Fig 1. *The Great Psalms Scroll*, 11Q5.** (A) Upper left: PAM infrared images with its 39 turns. (B) Lower left: IAA color images. (C) Right: closeup, showing wormholes. (Courtesy IAA, LLDSSD. IAA image photographer: Shai Halevi; PAM photographer: Najib Anton Albina.).

all, of the aforementioned problems with the PAM images. However, 50–70 years have passed since the imaging of the PAMs, during which time the physical artifacts have suffered additional damage, affecting the material reconstruction of the scrolls. (In fact, most of the shrinkage and other damage to the scrolls occurred already within the two millennia of their

(A)

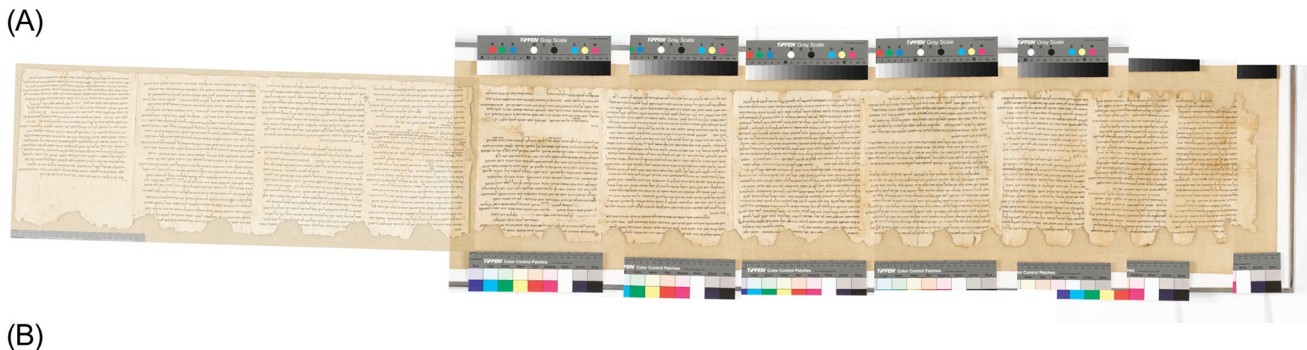

(B)

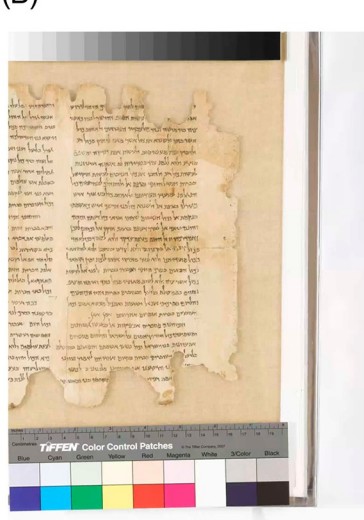

**Fig 2. _Community Rule,_ 1QS.** (A) Left: full scroll. (B) Right: closeup, showing damage patterns and ruler. (Courtesy Shrine of the Book, Israel Museum. Photographer: Ardon Bar-Hama.).

perdurance in caves, but it is not always easy to estimate these damages, and mostly we cannot overcome this factor. Herber [40, p. 8] estimates the shrinkage in some places as reaching up to 40% of the original width.) See Figs 6C and 1B for full-color examples of IAA images.

**Shrine of the Book images.** Photographer Ardon Bar-Hama imaged some of the large scrolls for the Shrine of the Book at the Israel Museum in Jerusalem in 2010–2011. The resolution of these images is wonderful, and the files include an industrial standard ruler. However, for some reason the files are not uniformly scaled. We, therefore, had to rescale each one individually in order to stitch them together. In addition, it is difficult to determine the extent of the material's shrinkage, since the older pictures do not usually include a ruler, which hampers any attempt at accurate measurements. See Fig 2A for an example.

## Measurements and calculations

Assumption C is that the distance between turns was constant throughout the rolled scroll. If we adopt a concentric circle model, then the circumferences of each turn will increase linearly, as would the distances between consecutive damages per Assumption B. However, even in cases where the damage pattern is clear, we lack a standardized procedure for measuring the distances between damages. Each turn of the scroll was damaged together with the other layers, but also continued to deteriorate in its own unique way. In addition, since the damage

itself has a certain width, the decision from which point to measure distances is oftentimes debatable.

For the purpose of measuring the actual distances between consecutive points and calculating their differences and the entire length of the scroll, we examined both PAM and IAA images of 11Q5 (= 11QPs) and of 11Q11 (= 11QApocPs), and new images of 1QS provided to us by the Shrine of the Book. After establishing the correct scale using GIMP (GNU Image Manipulation Program), we stitched the separate images together into a long continuous scroll using Adobe InDesign. We then found the repeating pattern of damages, and measured the horizontal distance between damage points. The results were gathered in a spreadsheet, in which we calculated the differences between these distances, and the length of the rest of the scroll from each point assuming only three or four damages were preserved. For some of the more complex calculations and tables, we coded Lisp programs. These results are compared to the measured length of that part of the scroll.

## Circle approximation

In this section, we derive the formulæ for determining the length of a scroll, imagining that it is instead a nested collection of cylindrical turns, as done in Stegemann [4] and Stoll [6]. See Fig 3A and 3B.

With three damage points, one can measure the horizontal distance between them and—relying on Assumption B—use that as a surrogate for the circumference of each of two turns. Let $c$ be the outermost circumference, and $b$ the next one just inside. The difference between the two, $z = c - b$, is, by Assumption C, the same for every turn. If there were $n$ turns to the scroll up to this point, then the total length $L$ (including these outermost two turns) is as follows:

$$
\begin{aligned}
L \quad &= c + (c - z) + (c - 2z) + \cdots + (c - (n-1)z) \\
&= nc - z(0 + 1 + \cdots + (n-1)) \\
&= nc - zn(n-1)/2
\end{aligned}
\tag{1}
$$

The last line is known by the famous "Gauss's formula,"

$$
0 + 1 + 2 + \cdots + (n-1) = n(n-1)/2
$$

for "triangular numbers." See Fig 4.

The above Eq (1) is identical to Stoll's formula (3), except that the latter is formulated in terms of the radius $r$ of the innermost or outermost circle.

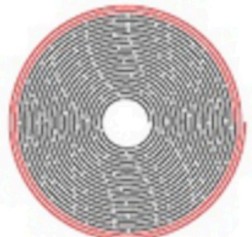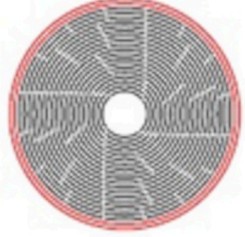

**Fig 3. Approximation of spiral by circles.** (A) Left: Spiral, with an increase in radius of $2\pi\rho = 0.1$ per turn. (B) Right: Concentric circles with radii $r = 0.5, 0.6, \ldots, 2.4, 2.5$. The red solid turns are preserved; the black broken ones are to be reconstructed.

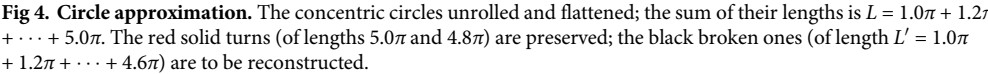

**Fig 4. Circle approximation.** The concentric circles unrolled and flattened; the sum of their lengths is $L = 1.0\pi + 1.2\pi + \cdots + 5.0\pi$. The red solid turns (of lengths $5.0\pi$ and $4.8\pi$) are preserved; the black broken ones (of length $L' = 1.0\pi + 1.2\pi + \cdots + 4.6\pi$) are to be reconstructed.

It remains to determine the number of turns $n$. Let's assume that $e$ is the circumference of the core, in which case $r_0 = e/(2\pi)$ is its radius. We need to choose $n$ so that the innermost turn (with circumference $c - (n - 1)z$) is not within the core, whereas with one more inner turn (of circumference $c - nz$) it would be inside. Accordingly, we need to choose the integer $n$ satisfying

$$c - nz < e \le c - (n - 1)z$$

which is the smallest $n$ such that

$$\frac{c - e}{z} < n$$

In other words,

$$n = \left\lfloor \frac{c - e}{z} \right\rfloor + 1 \tag{2}$$

where $\lfloor (c - e)/z \rfloor$ is $(c - e)/z$ rounded down.

To summarize, assuming $e$ and measuring $c$ and $b$, we proceed per the following steps (3–5):

$$z = c - b \tag{3}$$

$$n = \left\lfloor \frac{c - e}{z} \right\rfloor + 1 \tag{4}$$

$$L = n(c - z(n - 1)/2) \tag{5}$$

When $e = r_0 = 0$, that is, presuming the scroll had no rod at its core and is rolled all the way to the center, $n = \lfloor c/z \rfloor + 1$ (cf. Stoll's (7)), and formula (5) for length becomes

$$L = \left( \left\lfloor \frac{c}{z} \right\rfloor + 1 \right) \left( c - \frac{z}{2} \left\lfloor \frac{c}{z} \right\rfloor \right) \tag{6}$$

Based on this calculation, we have prepared Table 1, which can be used in lieu of the above formulæ to estimate the length of a scroll as a function of the quotient $c/z$, a value that is known from measurements (recall that $z = c - b$). The entries in the table are all in units of $z$. Take 11Q5 as an example. Worm holes 23 and 24 are 10.24 cm apart, while the distance between holes 24 and 25 is 10.65 cm. Taking $c = 10.65$ and $z = 10.65 - 10.24 = 0.41$, we derive $c/z \approx 26$, for which table lookup gives 351, corresponding to a scroll length of $351z$, or approximately 144 cm. Thus, the length of the scroll up to hole 23 should be 123 cm. The actual length, however, is about 136, a difference of almost 10% between the calculation and reality. Linear interpolation should be used for intermediate values.

**Table 1. Lookup table for the actual length of a spiral scroll.**

| c/z | L/z | c/z | L/z | c/z | L/z | c/z | L/z | c/z | L/z |
|-----|-----|-----|-----|-----|-----|-----|-----|-----|-----|
| 1 | 1 | 11 | 66 | 21 | 231 | 31 | 496 | 41 | 861 |
| 2 | 3 | 12 | 78 | 22 | 253 | 32 | 528 | 42 | 903 |
| 3 | 6 | 13 | 91 | 23 | 276 | 33 | 561 | 43 | 946 |
| 4 | 10 | 14 | 105 | 24 | 300 | 34 | 595 | 44 | 990 |
| 5 | 15 | 15 | 120 | 25 | 325 | 35 | 630 | 45 | 1035 |
| 6 | 21 | 16 | 136 | 26 | 351 | 36 | 666 | 46 | 1081 |
| 7 | 28 | 17 | 153 | 27 | 378 | 37 | 703 | 47 | 1128 |
| 8 | 36 | 18 | 171 | 28 | 406 | 38 | 741 | 48 | 1176 |
| 9 | 45 | 19 | 190 | 29 | 435 | 39 | 780 | 49 | 1225 |
| 10 | 55 | 20 | 210 | 30 | 465 | 40 | 820 | 50 | 1275 |

To use, measure gap $c$ and difference $z = c − b$ between $c$ and the next gap $b$ inwards. Lookup $c/z$ here and multiply the tabular value $L/z$ by $z$ to obtain the total length $L$ (including $b$ and $c$).

For example, if $c/z = 19.5$, then the length is midway between the entry 190 for 19 and 210 for 20, that is $L/z = 190 + (19.5 − 19) \times (210 − 190) = 200$, which needs to be multiplied by $z$ to obtain the scroll length $L$.

In the final analysis, it makes almost no difference whether $n = c/z$ is fractional, or rounded down as in (2). So, one could just use

$$L^* = \frac{c}{2}\left(\frac{c}{z} + 1\right) = \frac{c^2}{2z} + \frac{c}{2} \tag{7}$$

instead of (6). See Table 2.

The missing part of the scroll, excluding the outer two turns, which are the ones measured, is

$$L' = L − c − b = L − 2c + z \tag{8}$$

as depicted in Fig 5.

**n.b**. This is the formula used to compute the errors of estimation in Tables 3–6, using (5) for $L$, since we are interested in the degree of error in the estimate of the unknown part $L'$.

As an aside, Eq (7) can be inverted by applying the standard formula for solutions to a quadratic equation to the equation

$$\frac{z}{2}n^2 + \frac{z}{2}n − L = 0$$

where $n = c/z$. This allows one to approximate the number $n$ of turns from the presumed length $L$ ($L^*$ in Eq 7) and (positive) increment $z$, as follows:

$$n = \frac{-(z/2) \pm \sqrt{(z/2)^2 − 4(z/2)(−L)}}{2(z/2)} = \frac{1}{2}\sqrt{\frac{8L}{z} + 1} − \frac{1}{2} \approx \sqrt{\frac{L}{\pi s}} − \frac{1}{2} \tag{9}$$

where $s = z/(2\pi)$ is the increase in radius per turn. In our example, with $L/z = 200$, we get $n = 0.5 \times \sqrt{8 \times 200 + 1} − 0.5 \approx 19.5$, as expected. (Cf. Stoll's (5), which is the same except that it also accounts for a core.).

Not every scroll has a core, and when there is one its size can vary, so our estimates ignored that possibility. If one chooses to take a core of circumference $e$ and radius $r_0$ into account,

**Table 2. Comparison of methods of estimating the length of the missing part of a spiral scroll.**

| Turns $n$ | Length $L$ | Gap $c$ | Diff. $z$ | Ratio $c/z$ | Circle | | Wedge | Spiral | |
| | | | | | fractional | whole | | approximation | calculation |
|---|---|---|---|---|---|---|---|---|---|
| 1 | 3.38 | 3.38 | 3.38 | 1.00 | 0.00 | 3.22 | -0.52 | 3.46 | 3.38 |
| 2 | 12.86 | 9.48 | 6.10 | 1.55 | 9.47 | 12.66 | 7.29 | 12.96 | 12.86 |
| 3 | 28.60 | 15.74 | 6.26 | 2.51 | 25.19 | 28.37 | 21.40 | 28.70 | 28.60 |
| 4 | 50.62 | 22.01 | 6.27 | 3.51 | 47.20 | 50.37 | 41.80 | 50.72 | 50.62 |
| 5 | 78.91 | 28.29 | 6.28 | 4.51 | 75.48 | 78.65 | 68.50 | 79.02 | 78.91 |
| 6 | 113.48 | 34.57 | 6.28 | 5.51 | 110.04 | 113.21 | 101.48 | 113.59 | 113.48 |
| 7 | 154.33 | 40.85 | 6.28 | 6.50 | 150.89 | 154.05 | 140.74 | 154.44 | 154.33 |
| 8 | 201.47 | 47.13 | 6.28 | 7.50 | 198.01 | 201.17 | 186.29 | 201.58 | 201.47 |
| 9 | 254.89 | 53.42 | 6.28 | 8.50 | 251.42 | 254.58 | 238.12 | 255.00 | 254.88 |
| 10 | 314.58 | 59.70 | 6.28 | 9.50 | 311.12 | 314.27 | 296.24 | 314.70 | 314.58 |
| 11 | 380.56 | 65.98 | 6.28 | 10.50 | 377.09 | 380.25 | 360.64 | 380.68 | 380.56 |
| 12 | 452.83 | 72.26 | 6.28 | 11.50 | 449.35 | 452.50 | 431.33 | 452.94 | 452.83 |
| 13 | 531.37 | 78.55 | 6.28 | 12.50 | 527.89 | 531.04 | 508.29 | 531.49 | 531.37 |
| 14 | 616.20 | 84.83 | 6.28 | 13.50 | 612.71 | 615.87 | 591.54 | 616.32 | 616.20 |
| 15 | 707.32 | 91.11 | 6.28 | 14.50 | 703.82 | 706.97 | 681.08 | 707.43 | 707.31 |
| 16 | 804.71 | 97.39 | 6.28 | 15.50 | 801.21 | 804.36 | 776.90 | 804.82 | 804.71 |
| 17 | 908.39 | 103.68 | 6.28 | 16.50 | 904.88 | 908.03 | 879.00 | 908.50 | 908.39 |
| 18 | 1018.35 | 109.96 | 6.28 | 17.50 | 1014.84 | 1017.99 | 987.38 | 1018.46 | 1018.35 |
| 19 | 1134.59 | 116.24 | 6.28 | 18.50 | 1131.08 | 1134.23 | 1102.05 | 1134.70 | 1134.59 |
| 20 | 1257.12 | 122.53 | 6.28 | 19.50 | 1253.60 | 1256.75 | 1223.00 | 1257.23 | 1257.12 |
| 21 | 1385.93 | 128.81 | 6.28 | 20.50 | 1382.41 | 1385.56 | 1350.23 | 1386.04 | 1385.93 |
| 22 | 1521.02 | 135.09 | 6.28 | 21.50 | 1517.50 | 1520.65 | 1483.75 | 1521.13 | 1521.02 |
| 23 | 1662.39 | 141.38 | 6.28 | 22.50 | 1658.87 | 1662.02 | 1623.55 | 1662.51 | 1662.40 |
| 24 | 1810.05 | 147.66 | 6.28 | 23.50 | 1806.52 | 1809.67 | 1769.63 | 1810.17 | 1810.05 |
| 25 | 1963.99 | 153.94 | 6.28 | 24.50 | 1960.46 | 1963.61 | 1922.00 | 1964.11 | 1963.99 |
| 26 | 2124.22 | 160.22 | 6.28 | 25.50 | 2120.68 | 2123.83 | 2080.65 | 2124.33 | 2124.22 |
| 27 | 2290.72 | 166.51 | 6.28 | 26.50 | 2287.19 | 2290.34 | 2245.59 | 2290.84 | 2290.73 |
| 28 | 2463.51 | 172.79 | 6.28 | 27.50 | 2459.98 | 2463.13 | 2416.80 | 2463.64 | 2463.52 |
| 29 | 2642.59 | 179.07 | 6.28 | 28.50 | 2639.05 | 2642.20 | 2594.30 | 2642.71 | 2642.59 |
| 30 | 2827.95 | 185.36 | 6.28 | 29.50 | 2824.40 | 2827.55 | 2778.08 | 2828.06 | 2827.94 |
| 31 | 3019.58 | 191.64 | 6.28 | 30.50 | 3016.04 | 3019.19 | 2968.15 | 3019.70 | 3019.58 |
| 32 | 3217.51 | 197.92 | 6.28 | 31.50 | 3213.96 | 3217.11 | 3164.50 | 3217.63 | 3217.51 |
| 33 | 3421.71 | 204.21 | 6.28 | 32.50 | 3418.17 | 3421.31 | 3367.13 | 3421.83 | 3421.72 |
| 34 | 3632.20 | 210.49 | 6.28 | 33.50 | 3628.66 | 3631.80 | 3576.05 | 3632.32 | 3632.21 |
| 35 | 3848.97 | 216.77 | 6.28 | 34.50 | 3845.42 | 3848.57 | 3791.24 | 3849.09 | 3848.97 |
| 36 | 4072.03 | 223.06 | 6.28 | 35.50 | 4068.48 | 4071.62 | 4012.73 | 4072.15 | 4072.03 |
| 37 | 4301.37 | 229.34 | 6.28 | 36.50 | 4297.80 | 4300.95 | 4240.49 | 4301.48 | 4301.36 |
| 38 | 4536.99 | 235.62 | 6.28 | 37.50 | 4533.44 | 4536.58 | 4474.54 | 4537.11 | 4537.00 |
| 39 | 4778.90 | 241.90 | 6.28 | 38.50 | 4775.30 | 4778.44 | 4714.84 | 4778.97 | 4778.86 |
| 40 | 5027.08 | 248.19 | 6.28 | 39.50 | 5023.49 | 5026.63 | 4961.45 | 5027.16 | 5027.05 |

The table shows negligible differences between methods. Columns are: number $n$ of turns; length of spiral with $n$ turns and radius increase $s = 1$ between turns; length $c$ of outermost turn (i.e. gap between damage points); difference $z$ between outermost two turns; ratio $c/z \approx n$; concentric circle approximation for $c/z$ turns, per Eq (7) in Section S1 Appendix: Spiral Approximation in S1 File; concentric circle approximation for $\lfloor c/z \rfloor + 1$ turns per Eq (6); wedge approximation per Eq 23 in Section S2 Appendix: Wedge Approximation in S1 File; spiral approximation per Eq 13 of Section S1 Appendix: Spiral Approximation; precise calculation per Eq 15 of Section S1 Appendix: Spiral Approximation in S1 File. **n.b.** All these approximations are based on the values of the "measured" gaps $c$ from the next two rows. When the radius increases by $s \neq 1$, multiply tabulated values by $s$.

**Fig 5. Circle approximation estimation method.** The red solid turns (of lengths $c$ and $b = c − z$) are preserved; the black broken ones (of combined length $L'$) are to be reconstructed.

then just subtract

$$\frac{e}{2}\left(\frac{e}{z}+1\right) \quad = \pi r_0\left(\frac{2\pi r_0}{z}+1\right) \tag{10}$$

from (7) and from the length obtained using Table 1.

Since $\lfloor c/z \rfloor \leq c/z$, one can bound the overall length $L$ as

$$L \quad = \frac{c}{2}\left(\frac{c}{z}+1\right) + \frac{z\varepsilon}{2}(1-\varepsilon) \leq \frac{c}{2}\left(\frac{c}{z}+1\right) + \frac{z}{8} \tag{11}$$

where $\varepsilon = c/z \bmod 1$ is the fractional part of $c/z$, and $\varepsilon(1 − \varepsilon)$ takes its maximum at $\varepsilon = 1/2$ (and minimum at $\varepsilon = 0$). Stoll derives this inequation from the quadratic formula. His (6) has negative $z$, whereas $z$ is positive for us. It follows that

$$L^* \leq L \leq L^* + z/8$$

where $L^*$ is as in Eq (7).

If a preserved fragment has only two damage points, then $z$ is not known. All that is known is one complete circumference $c$. In such a case, one might measure the thickness of the material and based on some inkling how tightly the scroll was rolled, estimate the difference $s$ in radii between turns, deducing

$$z \quad = 2\pi s$$
$$b \quad = c - z$$

and continuing as above. The problem is that the physical fragments themselves are rarely made accessible, so their thickness cannot be measured, and in any case the thickness of the material can change over the years and need not be constant for the whole of a manuscript. Nor is the tightness of the winding likely to be even throughout, a problem that one faces in any event.

## Results

In what follows, we examine the validity of each of the above assumptions with respect to the few preserved scrolls that were discovered still rolled up.

While Assumption A, that the currently visible repeating patterns of damage were introduced while the scroll was still rolled, may be true for some scrolls, it is not always the case. For example, examining the new IAA images of 11Q10 (= 11QTgJob, *Targum Job*) in Fig 6C, the repeating pattern of damage is very clear. However, checking the older PAM images of the same scroll in Fig 6B, which were made as it was unrolled, reveals that much of this damage only materialized during the opening process, as pieces of leather crumbled. Indeed, examining the image taken while the scroll was still rolled in Fig 6A, shows that the repeating damage is not yet present. If the damages shown in the IAA images would have been created while the scroll was still rolled, we would have expected the upper edges of the scroll to be nearly straight

**Table 3.** *The Great Psalms Scroll*, 11Q5.

| | Damage Number $n$ | Actual Length $L$ | PAM | | | | IAA | | | |
|---|---|---|---|---|---|---|---|---|---|---|
| | | | Gap $c$ | Diff. $z$ | Error % 3 points | Error % 4 points | Gap $c$ | Diff. $z$ | Error % 3 points | Error % 4 points |
| 1 | 0–1 | 4.15 | 4.15 | | 80 | 44 | 4.15 | | 52 | 42 |
| 2 | 1–2 | 6.62 | 2.47 | | 0 | 33 | 2.45 | | 16 | 32 |
| 3 | 2–3 | 9.44 | 2.82 | 0.35 | 91 | 103 | 2.85 | 0.40 | 48 | 60 |
| 4 | 3–4 | 12.76 | 3.32 | 0.50 | 89 | 47 | 3.30 | 0.45 | 60 | 47 |
| 5 | 4–5 | 16.36 | 3.60 | 0.28 | 8 | -9 | 3.65 | 0.35 | 24 | 24 |
| 6 | 5–6 | 20.21 | 3.85 | 0.25 | -22 | **− 22** | 3.95 | 0.30 | 20 | 72 |
| 7 | 6–7 | 24.44 | 4.23 | 0.38 | **525** | **− 3** | 4.30 | 0.35 | 185 | 68 |
| 8 | 7–8 | 29.17 | 4.73 | 0.50 | -21 | -10 | 4.65 | 0.35 | 5 | -3 |
| 9 | 8–9 | 33.83 | 4.66 | **− 0.07** | 9 | 16 | 4.80 | 0.15 | -10 | 4 |
| 10 | 9–10 | 38.92 | 5.09 | 0.43 | 23 | 41 | 5.15 | 0.35 | 25 | 37 |
| 11 | 10–11 | 44.34 | 5.42 | 0.33 | 61 | 38 | 5.55 | 0.40 | 48 | 65 |
| 12 | 11–12 | 50.05 | 5.71 | 0.29 | 15 | 2 | 5.85 | 0.30 | 80 | 18 |
| 13 | 12–13 | 55.98 | 5.93 | 0.22 | -10 | -28 | 6.10 | 0.25 | -17 | -15 |
| 14 | 13–14 | 62.20 | 6.22 | 0.29 | -40 | 5 | 6.30 | 0.20 | -10 | -22 |
| 15 | 14–15 | 68.78 | 6.58 | 0.36 | 262 | -19 | 6.70 | 0.40 | -31 | -16 |
| 16 | 15–16 | 75.89 | 7.11 | 0.53 | -59 | -28 | 7.08 | 0.38 | 9 | 25 |
| 17 | 16–17 | 83.10 | 7.21 | 0.10 | 168 | 48 | 7.57 | 0.49 | 45 | 29 |
| 18 | 17–18 | 91.06 | 7.96 | 0.75 | -4 | -26 | 7.90 | 0.33 | 13 | 1 |
| 19 | 18–19 | 99.16 | 8.10 | 0.14 | -40 | -26 | 8.15 | 0.25 | -9 | 0 |
| 20 | 19–20 | 107.62 | 8.46 | 0.36 | 1 | -8 | 8.46 | 0.31 | 13 | -9 |
| 21 | 20–21 | 116.64 | 9.02 | 0.56 | -16 | -16 | 8.84 | 0.38 | -24 | -28 |
| 22 | 21–22 | 126.02 | 9.38 | 0.36 | -15 | -13 | 9.15 | 0.31 | -31 | -18 |
| 23 | 22–23 | 135.83 | 9.81 | 0.43 | -10 | -15 | 9.60 | 0.45 | 4 | -15 |
| 24 | 23–24 | 146.07 | 10.24 | 0.43 | -19 | -12 | 10.10 | 0.50 | -29 | 3 |
| 25 | 24–25 | 156.72 | 10.65 | 0.41 | -3 | 32 | 10.45 | 0.35 | 88 | 88 |
| 26 | 25–26 | 167.83 | 11.11 | 0.46 | 104 | -15 | 10.95 | 0.50 | 82 | 61 |
| 27 | 26–27 | 179.33 | 11.50 | 0.39 | -49 | -25 | 11.15 | 0.20 | 41 | -1 |
| 28 | 27–28 | 191.02 | 11.69 | 0.19 | 40 | 51 | 11.35 | 0.20 | -25 | -20 |
| 29 | 28–29 | 203.41 | 12.39 | 0.70 | 61 | 68 | 11.60 | 0.25 | -14 | 27 |
| 30 | 29–30 | 216.08 | 12.67 | 0.28 | 72 | 10 | 12.05 | 0.45 | 136 | 7 |
| 31 | 30–31 | 228.99 | 12.91 | 0.24 | -21 | -27 | 12.45 | 0.40 | -33 | -21 |
| 32 | 31–32 | 242.12 | 13.13 | 0.22 | -31 | -35 | 12.60 | 0.15 | -1 | -28 |
| 33 | 32–33 | 255.71 | 13.59 | 0.46 | -37 | | 13.10 | 0.50 | -44 | |
| 34 | 33–34 | 269.83 | 14.12 | 0.53 | | | 13.45 | 0.35 | | |
| 35 | 34–35 | 284.54 | 14.71 | 0.59 | | | 14.05 | 0.60 | | |

Measured and calculated lengths based on three and four consecutive points using PAM and IAA images of 11Q5. The measurements are in cm. Gaps $c$ between damage points are measured. Differences $z$ are the changes in distance between successive gaps. All errors are for the circle approximation (based on $c$ and $z$) of the length of the remainder of the scroll, inwards from the innermost of the measured damage points (Eq 8 for $L'$ in Section Circle Approximation); they are given as percentages with respect to the measured PAM length $L$. **Emboldened** values are due to a negative difference, which is ignored for 4 points, when a positive difference is also available.

and the bottom edges to be lower at one side and higher at the other, as to reflect the observed damage of the open scroll. In fact in both sides most of the edges are nearly straight.

In the case of 11Q10, we have documentation for the introduction of relevant damages *while* the scroll was being unrolled. This means that at least some of the visible damage may have been precipitated during the unrolling of the scroll. Most scrolls, however, came into our

**Table 4.** *Apocryphal Psalms*, **11Q11.**

| | Damage Number $n$ | Actual Length $L$ | PAM | | | | IAA | | | |
|---|---|---|---|---|---|---|---|---|---|---|
| | | | Gap $c$ | Diff. $z$ | Error % 3 points | Error % 4 points | Gap $c$ | Diff. $z$ | Error % 3 points | Error % 4 points |
| 1 | 1–2 | 1.85 | 1.85 | | 227 | 274 | 1.70 | | 482 | 462 |
| 2 | 2–3 | 4.56 | 2.71 | 0.86 | 155 | 15 | 2.83 | 1.13 | 178 | **178** |
| 3 | 3–4 | 7.77 | 3.21 | 0.50 | -48 | − **48** | 3.16 | 0.33 | **232** | − **81** |
| 4 | 4–5 | 11.37 | 3.60 | 0.39 | **3117** | 178 | 3.51 | 0.35 | -89 | − **89** |
| 5 | 5–6 | 16.10 | 4.73 | 1.13 | 93 | 140 | 3.31 | − **0.20** | − **67** | − **52** |
| 6 | 6–7 | 20.80 | 4.70 | − **0.03** | 178 | 62 | 5.35 | 2.04 | -80 | -74 |
| 7 | 7–8 | 25.83 | 5.03 | 0.33 | -1 | 1 | 4.16 | − **1.19** | -49 | − **49** |
| 8 | 8–9 | 31.07 | 5.24 | 0.21 | 3 | 49 | 5.54 | 1.38 | **6** | **2156** |
| 9 | 9–10 | 36.80 | 5.73 | 0.49 | 153 | **153** | 6.50 | 0.96 | 1522 | **1522** |
| 10 | 10–11 | 43.00 | 6.20 | 0.47 | **69** | 51 | 6.00 | − **0.50** | 205 | − **22** |
| 11 | 11–12 | 49.40 | 6.40 | 0.20 | 21 | | 6.03 | 0.03 | -35 | |
| 12 | 12–13 | 55.55 | 6.15 | − **0.25** | | | 5.90 | − **0.13** | | |
| 13 | 13–14 | 62.00 | 6.45 | 0.30 | | | 6.40 | 0.50 | | |

Measured and calculated lengths based on three and four consecutive points using PAM and IAA images of 11Q11. (See legend at Table 3.)

hands fragmented and not rolled. Once detached from the original scroll, each fragment deteriorated separately. It is not, therefore, always easy to tell when and how each crack or tear originated.

Even in cases where Assumption A can be accepted as reasonable, Assumption B, viz. that measuring damage points yields the circumference, needs to be carefully evaluated. For example, in 1QS (*Serekh haYahad*, *Community Rule*, formerly *Manual of Discipline*), two patterns of damage are simultaneously evident, one along the top margin and the other on the bottom; see Fig 2. There are approximately double the number of damages on the top than on the bottom, which probably indicates that the top was damaged in two separate places, while the bottom in only one. Thus, the circumference of the scroll at a certain point should not be measured between two consecutive upper damage points, but rather between every other damage. However, had only the top margin of 1QS been preserved, we would have had no way of knowing this since the difference between the two patterns is not obvious. Thus, our estimation of the circumference would have been approximately half the true length. Since very few scrolls have survived comparatively intact, it is difficult to assess how common or rare this situation, with multiple patterns of damages that look as if they belong to only one pattern, might be overall.

**Table 5.** *Apocryphal Psalms*, **11Q11.**

| Damage | $n$ | 1-2 | 2-3 | 3-4 | 4-5 | 5-6 | 6-7 | 7-8 | 8-9 | 9-10 | 10-11 | 11-12 |
|---|---|---|---|---|---|---|---|---|---|---|---|---|
| Actual | $L$ | 1.85 | 4.56 | 7.77 | 11.37 | 16.10 | 20.80 | 25.83 | 31.07 | 36.80 | 43.00 | 49.40 |
| No | Estimate | 6.05 | 11.64 | 4.02 | −365.82 | 31.15 | 57.15 | 25.45 | 32.10 | 93.00 | −72.60 | 60.00 |
| Core | Error | 227% | 155% | −48% | * | 93% | 178% | −1% | 3% | 153% | * | 21% |
| 5 mm | Estimate | 3.92 | 8.94 | 2.47 | −325.00 | 27.45 | 51.92 | 22.96 | 28.92 | 87.40 | −67.95 | 56.25 |
| Core | Error | 112% | 96% | −68% | * | 70% | 150% | −11% | −7% | 137% | * | 14% |

Effect on estimated length of 11Q11, based on three PAM points, assuming a core of diameter 5 mm. Negative lengths indicate that the method fails to infer the correct direction.

**Table 6.** *Community Rule*, 1QS.

| | Damage Number $n$ | Actual Length $L$ | Shrine of the Book | | | |
|---|---|---|---|---|---|---|
| | | | Gap $c$ | Diff. $z$ | Error % 3 points | Error % 4 points |
| 1 | 0–1 | 2.40 | 2.40 | | 1887 | 1725 |
| 2 | 1–2 | 7.90 | 5.50 | | 472 | 638 |
| 3 | 2–3 | 13.70 | 5.80 | 0.30 | 568 | 430 |
| 4 | 3–4 | 19.85 | 6.15 | 0.35 | 223 | 197 |
| 5 | 4–5 | 26.20 | 6.35 | 0.20 | 128 | 263 |
| 6 | 5–6 | 32.85 | 6.65 | 0.30 | 635 | 119 |
| 7 | 6–7 | 39.85 | 7.00 | 0.35 | 6 | 18 |
| 8 | 7–8 | 46.95 | 7.10 | 0.10 | 30 | **30** |
| 9 | 8–9 | 54.60 | 7.65 | 0.55 | **279** | **64** |
| 10 | 9–10 | 62.70 | 8.10 | 0.45 | 38 | 28 |
| 11 | 10–11 | 70.65 | 7.95 | **− 0.15** | 16 | 89 |
| 12 | 11–12 | 78.95 | 8.30 | 0.35 | 374 | 86 |
| 13 | 12–13 | 87.65 | 8.70 | 0.40 | 5 | 91 |
| 14 | 13–14 | 96.45 | 8.80 | 0.10 | 773 | -28 |
| 15 | 14–15 | 105.65 | 9.20 | 0.40 | -67 | **67** |
| 16 | 15–16 | 114.90 | 9.25 | 0.05 | **42** | **51** |
| 17 | 16–17 | 125.25 | 10.35 | 1.10 | 30 | 30 |
| 18 | 17–18 | 135.30 | 10.05 | **− 0.30** | 28 | 35 |
| 19 | 18–19 | 145.65 | 10.35 | 0.30 | 41 | -9 |
| 20 | 19–20 | 156.30 | 10.65 | 0.30 | -34 | |
| 21 | 20–21 | 167.22 | 10.92 | 0.27 | | |
| 22 | 21–22 | 178.69 | 11.47 | 0.55 | | |

Measured and calculated lengths based on three and four consecutive points using Shrine of the Book images of 1QS. (See legend at Table 3.)

In [37, p. 109], another problem, caused by the oval profile of some of the scrolls with no core inside, is mentioned [41, p. 19]. In [36] mention is made of yet another problem that may be caused by an object penetrating the papyrus at an angle, thus creating a repeating pattern of damage that does not correspond to the circumference. (In a personal conversation, Annette Steudel offered a way to overcome this problem, namely, by requiring at least three examples of similar damages between every two layers.).

## 11Q5 (=11QPs, *The Great Psalms scroll*)

The Great Psalms Scroll, designated 11Q5 or 11QPs, was found by Bedouins and brought to the Palestinian Archaeological Museum in February 1956. It remained unrolled until November 1961.

11Q5 has two main merits for our purposes:

1. Its top half has remained almost intact, leaving more than 3 meters of continuous parchment; see Fig 1A.

2. In addition to the more common repeating pattern of water damage, which is to be found on most long scrolls, it also contains worm holes. Worm holes—unlike most other damage patterns—are very small, leaving merely a half a millimeter of possible measuring error;see Fig 1B.

(A)

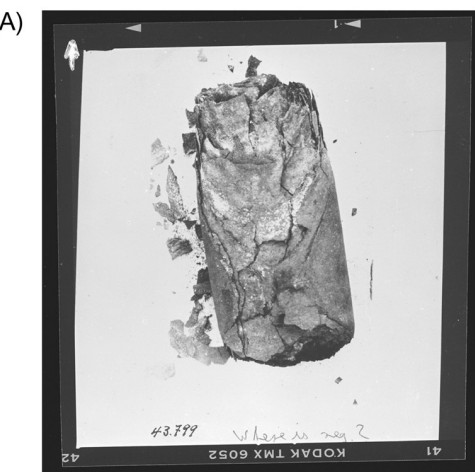

(B)

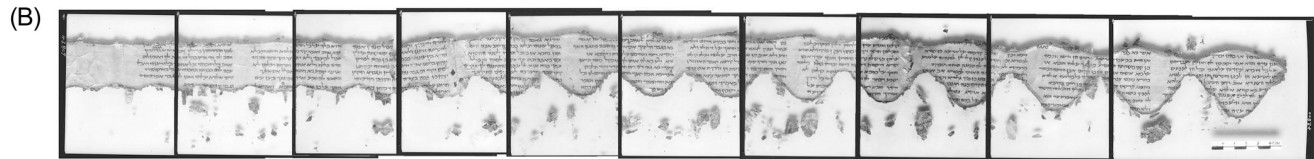

(C)

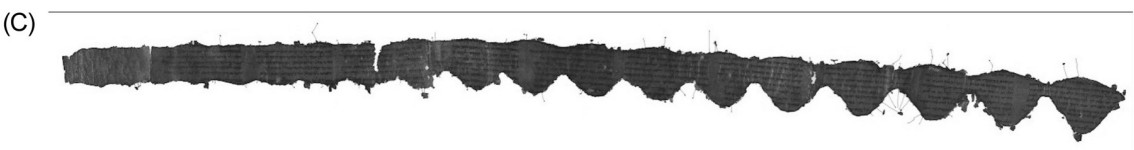

**Fig 6.** *Targum Job*, **11Q10.** (A) Upper left: prior to unrolling. (B) Upper right: composite of PAM images. (C) Lower right: composite of IAA images. (Courtesy IAA, LLDSSD. IAA image photographer: Shai Halevi; PAM photographer: Najib Anton Albina.).

The continuous part of the Great Psalms Scroll comprises three whole sheets, plus a fourth at the end. The scroll contains traces of 33 small holes, presumably made by a worm working its way from the outer part of the scroll through each of its layers to the core while the scroll was still rolled. (Stegemann [4, p. 212] states that the worm entered from the inside and made its exit via the bottom of the scroll around column 14. However, the worm holes do continue to the outermost columns of the preserved continuous part of the scroll. Though it is very difficult to discern them on the old PAM images, they are clearly visible on the new IAA color images. On the other hand, there is no worm hole in the innermost turn of the scroll.) The measured distances between every pair of consecutive holes is in fact the circumference of the scroll at that point.

PAM photos were taken during the unrolling process in November 10–20, 1961. Due to the length of the scroll, its continuous part was imaged in 12 separate photographs. As discussed above, like many other PAM images, the scaling of these images is not uniform. On each plate, next to the scroll, a hand-drawn ruler is present. The ruler's accuracy is unknown. While it is probable that the photographer used the same ruler in every photo, for some other reason each image has slightly different scaling, and the accuracy of none of them is certain. Before measuring the distances, we rescaled each image based on the hand-drawn ruler using GIMP (GNU Image Manipulation Program), and compared them to each other. An additional problem

with the PAM images is that, when imaged, the scroll was placed on a standard plate that was much shorter than the scroll, forcing the rest of the scroll to slide over onto the table. The resultant differences of height distort the edges of the images.

Unlike the old PAM images, scaling is not a problem with the new IAA images. These were taken in 2015 next to a manufactured ruler, with the camera set at a permanent position, planned in advance to create 1:1 scaling. However, apparently, over the years the scroll underwent shrinkage. Sanders [42] measured the length of the sheets (last one first) to be 77, 72, 87, and 81 cm, while—according to our measurements based on the 2015 images—they are now 75.5, 69.3, 83.0, and 72.4 cm, respectively. Since the shrinkage is not constant over the length of the scroll, it seems that, despite the better scaling, reconstructing the length of the original scroll based on the images of its current state is also problematic. (Obviously, shrinkage and other damaging processes also occurred during the two millennia prior to the unrolling of the scroll, but these cannot be measured today.) We began by measuring the distances between consecutive worm holes. Then we considered individual segments, each comprising three worm holes. For each such segment, we calculated the length of the scroll from that point inwards, based on the distances between the holes, imagining that this segment was the only preserved fragment. We then compared the estimates of the length of the scroll from that point inwards with measurements of the older PAM images taken before the subsequent shrinkage. In addition to the approximation based on concentric circles used previously by scholars, we also considered the slightly more accurate spiral approximation (for which, see above Circle approximation and below Supporting information).

Table 3 shows the measured length of the scroll (col. 2), the measured distances between damages for both PAM and IAA images (cols. 3 and 7), the difference between consecutive distances (cols. 4 and 8), and the error in percentage between the measured and calculated length of the scroll. The calculated length for each line is based on the measured distances given in the next two lines, since we calculate the length of the scroll from each damage inwards. As can be deduced from Table 3, Assumption C, namely, that the distances between points grow linearly, is not matched by reality. For the PAM images the difference ($z$) varies between $-0.07$ and 0.75 cm. The negative difference between successive circumferences is quite disconcerting. It means that, had only a fragment with these three points survived, one might have naïvely concluded that the scroll was rolled backwards, which is not the case. The average change in distance is 0.37 cm, and the standard deviation is 0.17 cm. Only 16 out of 33 (PAM: 3 points) estimates are within 25% of the true length. For the IAA images the situation is better in this regard. The difference varies between 0.15 and 0.6 cm, with no negative values, with an average difference of 0.35 cm and standard deviation 0.11 cm. Based on Stegemann's measurements [4], the average difference in consecutive distances between damage points in the entire Qumran corpus varies between 0.1 and 0.5 cm. Thus, the variance here in one manuscript alone is quite high. Fig 7 shows graphically the large fluctuations in estimates depending on which points are measured.

In the worst-case scenario (were only holes 7–9 preserved, lines 8–9 in the table), the measurements themselves give a negative difference between distances (line 9) instead of the correct, positive one. In such a case, the entire reconstruction would have assumed that the scroll was rolled with the beginning of the text at the inside end of the scroll; the resultant error is huge (line 7). In this particular case, we can explain away the problem, because the image is clearly distorted. However, this problem can occur on other occasions when the skin shrunk in some places but left no indication of the shrinkage.

Even if we exclude that one particular instance, the second-worst example (holes 15–17, lines 16–17) yields an error of over 250% (line 15). The cause for this may be the stitches between holes 15 and 16 and the start of the separation of these two sheets. It is very difficult to

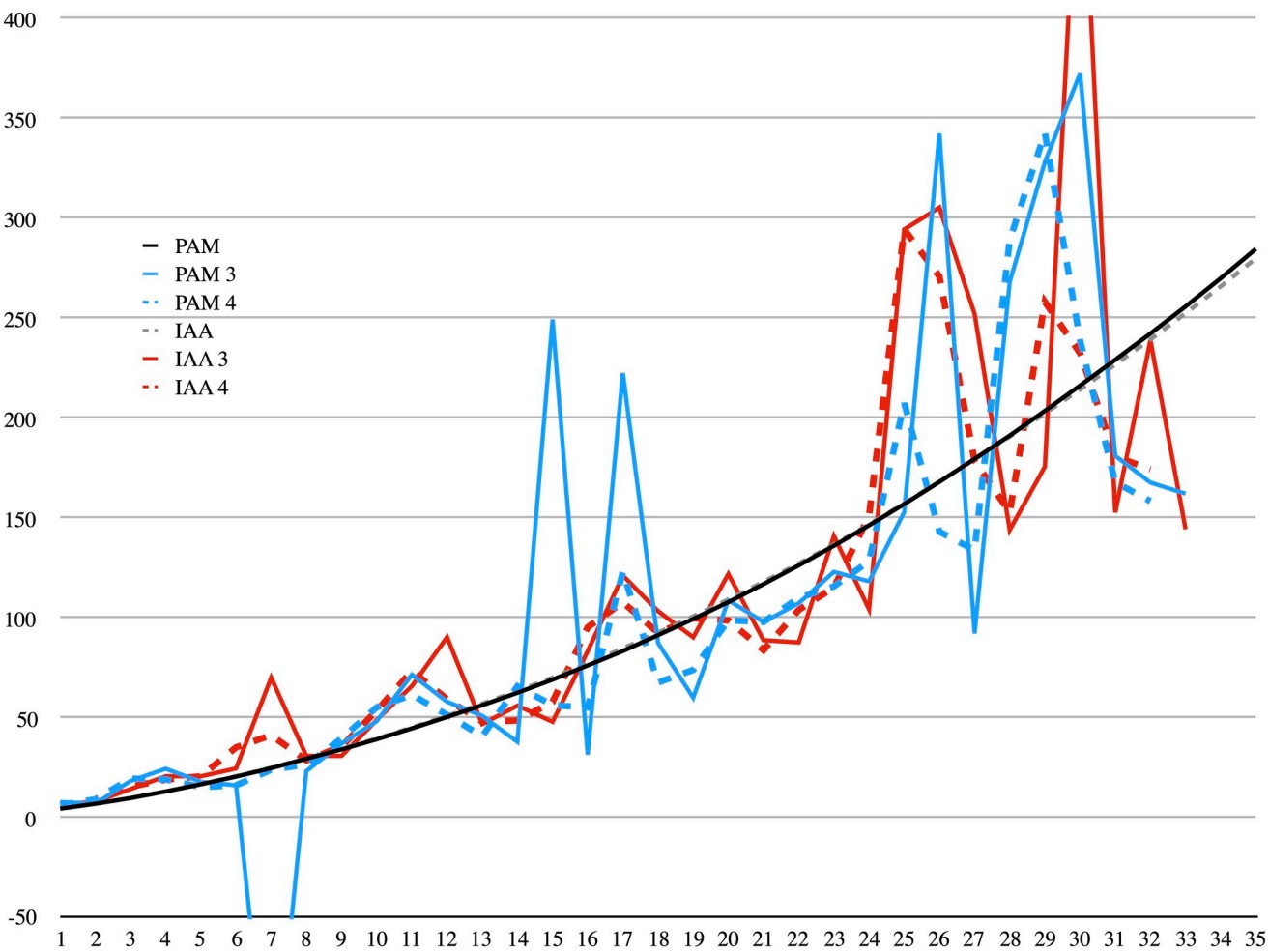

**Fig 7. 11Q5 (*The Great Psalms Scroll*).** Measured length compared to calculated lengths in cms. The smooth (black) curves are the actual lengths as measured on the PAM (solid) and IAA/Shrine (dashed) images. Calculated estimates are shown for both PAM (blue) and IAA/Shrine (red) fragments, for three damage points (solid) and four (dashed). Some outliers are "off the chart." A negative length indicates that the method fails to infer the correct direction of rolling.

avoid this kind of problem. In actual cases where the method has been applied, the state of the fragments was much worse, as separation of sheets or fragments is very frequent and hard to restore with high accuracy. Were the preserved holes 17–19 (lines 18–19), we would have been led to an error of around 170% (line 17). In this case, the problem is caused by the distortion of the picture. Were the preserved holes 26–28 (lines 27–28), the error would also have been over 100% (line 26), maybe because the scroll is not lying straight on the plate near hole 26. All the rest of the errors are below 100%. The average (absolute value of the) error, ignoring the one negative-difference case, is 46% with a standard deviation of 54%.

The same process, done on the new IAA images, gives better results. Had the preserved holes been 7–9 (lines 8–9), we would have an error of 185% (line 7). Holes 30–32 give an error of 136% (line 30). Holes 12–14, 25–27, and 26–28 give errors of 80–90% (lines 12, 25, 26). All the rest of the sections give significantly lower errors. The average error is 38%, with a standard deviation of 39%.

Had four, rather than only three, consecutive points of damage been preserved—in which case an average difference can be used in the calculations, this would have resulted in a

significantly reduced error. Moreover, if we assume that it is an experienced scholar who examines the scroll, and in case of one negative difference and one positive, she would choose the correct direction of rolling, the maximal error decreases to 100% and the average to 27% for the PAM images. The results are comparable for the IAA images, with no negative differences, a maximum error below 90%, and average around 28%. See Fig 7. Five consecutive points would further improve the estimates slightly. Table 3, as well as subsequent Tables (4) and (6), assume that in all such cases of conflicting differences, the correct direction is chosen.

## 11Q11 (=11QApocPs, *Apocryphal Psalms*)

Worm holes, as found in 11Q5, are very helpful for applying length-approximation methods; they limit much of the uncertainty and leave less room for measurement error. Unfortunately, scrolls with worm holes are very rare. In most cases, one has to rely on the human factor for deciding the exact points from which and to which one measures the gaps between damages. As an example of this more common situation, we examined 11Q11. For our purpose, this scroll has two merits:

1. An image of the scroll while still rolled is available (PAM 43.981). In both this image and the images of the opened scroll, the pattern of damage seems similar.

2. The shape of the damages facilitates the choice of the points from and to which measurements should be made.

Manuscript 11Q11 was acquired while still rolled in the shape of a small cigar measuring 8.5 cm height with a diameter of 3.2 cm. In order to open it, the scroll was exposed to humidified air for 30 minutes, which means that a change of shape might have occurred already in the earliest stages of its study, even before the first PAM images were taken in 1961. The length of the scroll was 73 cm when first measured [43]. In 1998, its length was only 71 cm, and we measure the same length today. The preserved scroll was written on a single sheet that was stitched to another blank one as a handle sheet. In the innermost part of the scroll 3–4 turns are still rolled and tied with a string. It was found with the handle rod still at the core, its diameter measuring 3–5 mm [20, p. 182].

When opened, 11Q11 presented a zigzag damage pattern, containing 15 peaks. As can be seen from the image of the unrolled scroll in Fig 8, the distance between every pair of consecutive peaks is the circumference of the scroll at that place ±0.5 cm. The scroll was torn between peaks 6–7, and we had to reconstruct it based on older images.

Scroll 11Q11 is documented on PAMs 43.982–8. Of them, only 43.985 includes a ruler. Thus, we had to assume that all photographs were taken on the same day, and rescaled them all based on the information that we had from that one with the ruler. In fact, the contents of every pair of images were somewhat overlapping. The fact that we were able to exactly match the overlapping parts proves our scaling decision to be correct.

The scroll is also spread over seven IAA images. Unlike the case of 11Q5, scaling here was slightly problematic, since not all images contain a ruler. Measuring pixels of the 4 cm rulers that are visible gave a range of 674–677 mm with respect to the image dpi.

So we chose an in-between measurement of 675.2 mm to rescale all of them uniformly. Again we were able to match the overlapping parts.

For the measurements of 11Q11, we used the same methods as for 11Q5. We measured the distances between proximate points of maximal height along the scroll. These points were chosen based on the image of the scroll while it was still rolled. No other repeating damage clearly correlated to the circumference of the scroll at that area. Since these maximal points varied in height, we use the horizontal component of the distance between points in our calculations.

(A) 11Q11

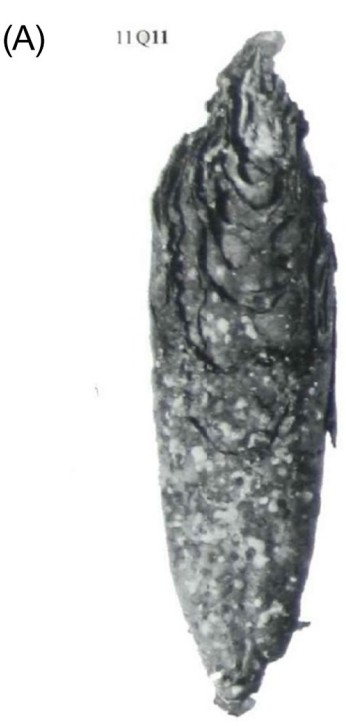

(B)

(C)

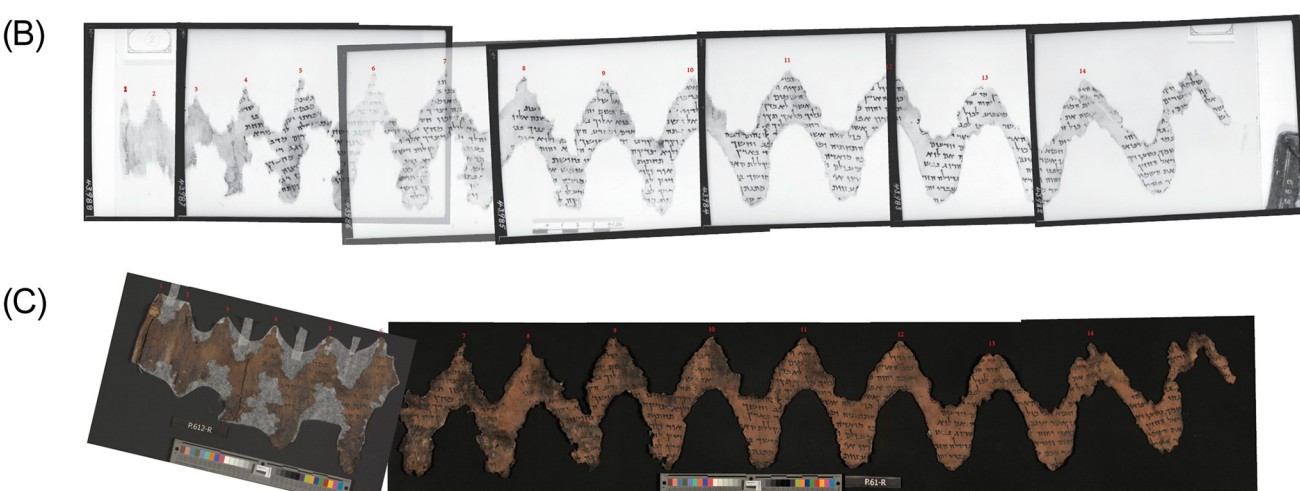

**Fig 8. *Apocryphal Psalms*, 11Q11.** (A) Left: prior to unrolling. (B) Upper right: composite of PAM infrared images. (C) Lower right: composite of IAA color images. (Courtesy IAA, LLDSSD. IAA image photographer: Shai Halevi; PAM photographer: Najib Anton Albina.).

Here too Assumption C need not hold. See Table 4 and Fig 9. The differences range from −0.25 mm to 1.13 mm. While the average difference of distances between damage points is 0.38, the standard deviation is nearly as large (0.36) for the PAM images, and twice as large for the IAA images (0.39 and 0.89, respectively). Two out of 13 differences in the PAM images were negative and 4 out of 13 in the IAA images. This means that, had only those points had been preserved, one would have concluded that the scroll was rolled with its beginning inside, and the calculations would be off 30-fold. Some of the errors are caused by the tear of the

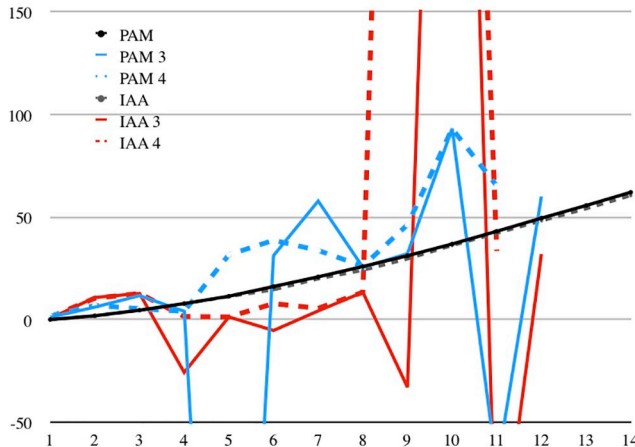

**Fig 9. 11Q11 (*Apocryphal Psalms*).** Measured length compared to calculated lengths in cms. See legend at Fig 7.

scroll, others by the difficulty of finding the exact points at which to measure, and sometimes the problem is shrinkage.

All these problems are unavoidable and show up in many scrolls. Only 3 out of 11 estimates for the PAMs are within 25% of the true length.

For this scroll, with four consecutive points preserved, the results are similar. In contrast to 11Q5, in this case the later IAA images were often worse for length-estimation purposes than the PAMs. See Fig 9.

Five consecutive points would be no better, and the hazard remains that one might mistake the scroll as being rolled the opposite way. On the other hand, in this case, taking a possible core into account would have a noticeable impact; see Table 5. Unfortunately, in actual cases, one usually has no way of knowing whether there was a core, and, if so, what its diameter was.

## 1QS (=Community Rule)

Since the error difference between the two above examples is significant, with 11Q5 proving to provide much more reliable estimates than 11Q11, we performed similar calculations also for 1QS. This scroll is one of the first seven scrolls found in 1947 in Cave 1, and is held at the Shrine of the Book. It was rolled with its beginning on the inside. As mentioned above, its innermost part displays two sets of damages in the upper and lower margins. There are twice the damage points in the upper margins than in the lower, thus indicating that the upper part of the scroll was damaged in two places, while its lower part in only one. We measured the distances between the maximum point of the damages on the bottom. See Fig 2A and 2B.

The scroll was imaged on seven separate plates, whose scaling is not uniform. After rescaling each one individually, it was possible to match them to each other, except for the images of columns 4 and 5. The stitches between the third and fourth sheets have deteriorated, and the two sheets have separated. We attempted to digitally rejoin them, but the possible error is accordingly large. See Fig 2.

Also in the case of 1QS, the difference between damage distances is not constant and varies between −0.3 cm and 1.1 cm with an average of 0.3 and a standard deviation of 0.28. See Table 6 and Fig 10. On two occasions the difference turns out to be negative (lines 11 and 18), which—as before—would falsely indicate that the scroll was rolled in the normal way with the beginning of the text at the outer end. As for the calculation of the length of the scroll, in the

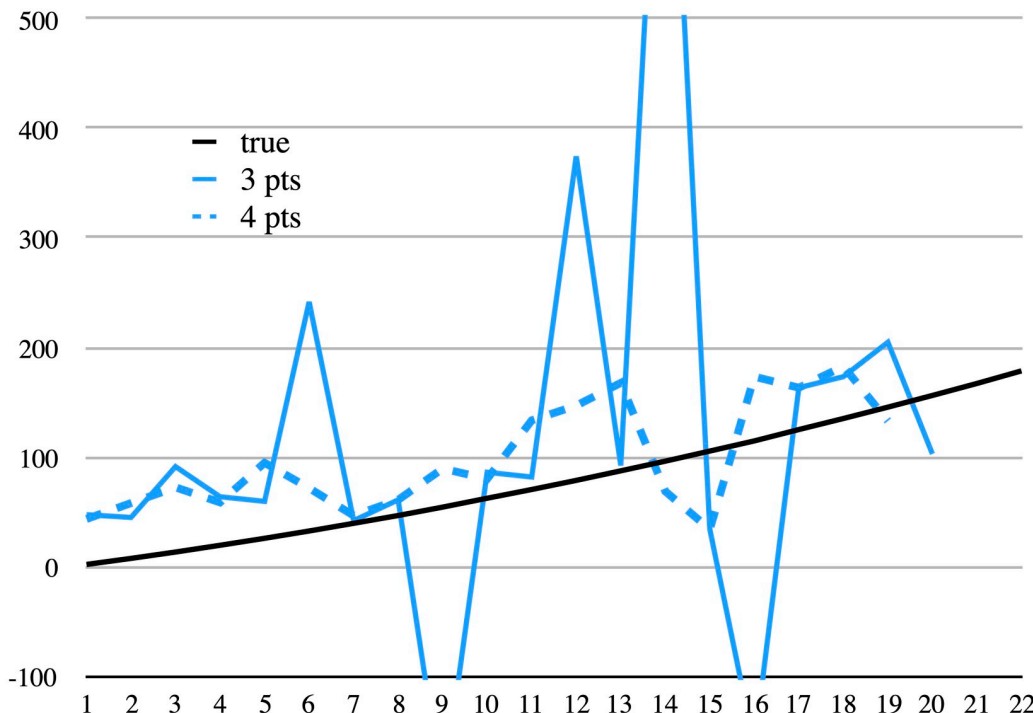

**Fig 10. 1QS (*Community Rule*).** Measured length compared to calculated lengths in cms. Calculated estimates are shown for three damage points (solid) and four (dashed). See legend at Fig 7.

worst case (were only damage points 1–3 preserved, lines 2–3), the error is over 1800%. This huge error may be explained by the fact that the first damage is quite different from the rest of the damages, and it is difficult to find the right point from which to measure the distance to the next damage. Another possibility is that scrolls rolled in the opposite direction, with the beginning to the inside, were rolled more loosely with a large core in the middle. Excluding the first error, if only points 14–16 were preserved (lines 15–16), the largest error comes out to approximately 770%. In nearly half the cases, the error is over 100%, and the average error (excluding the first one) is around 240% with a standard deviation of approximately 300%. Only 3 out of 20 estimates are within 25% of the true length. The scaling problems and separation of sheets may explain some of the errors, but it seems that the largest contribution comes from the subjectivity of the decision from where and to where to measure the distances between damage points.

Were four consecutive damage points to have survived, and presuming that in the case of one negative difference and one positive, we always are smart enough to choose the correct direction, this would ameliorate the errors somewhat, but they are still very large. See Table 6. For damage points 17–19, which annul each other, yielding an average difference of zero, the estimate must be made on the basis of only 3 of the points. (Fortunately, we encounter no instances of two negative values in a row.) See Fig 10.

Surprisingly, were five consecutive damage points to have survived, the results would have been somewhat worse.

## Discussion

All of the measurements of the previous section lead to one conclusion: While the length-reconstruction method works well in theory, its margin of error—when trying to estimate the

length of a real scroll—is simply too large to be trusted. In the rare cases where four or five consecutive damage points are preserved, however, its reliability might be reasonable. It should be stressed that we examined here some of the very best preserved scrolls, for which approximate reconstructions are not required, since they are already comparatively intact, and their length can be measured. In practice, the method is applied to much more poorly preserved scrolls, ones that have decomposed into many scattered fragments. These fragments underwent additional processes of deterioration, disintegration, and warping. In these cases, the distance between points is sometimes reconstructed rather than measured. Therefore, in the actual cases where the method is used, the margin of error should be expected to be much larger.

To summarize the results of our analyses, the reconstruction method faces numerous practical difficulties:

1. Most scrolls were probably not stored evenly rolled, so the assumption that consecutive turns are evenly spaced need not hold.

2. Repeating damage patterns are never perfect, so estimates can never be precise. In general, wormholes give considerably more accurate measurements than does water damage. Other means of improving the accuracy of measurements can also be devised. For example, one can use imprints of ink from one layer that appear on the verso of the adjacent one. For an instance of this phenomenon, see [18]; however, the fragments were quite small in that case, and no instances of three consecutive damages have been preserved.

3. Choosing the right points from which to measure—especially for instances of water damage along the edges—is an art and a challenge. Choosing less than ideal points leads to additional errors. (It has been suggested to use continuous autocorrelation as an objective, mathematical means of identifying the repeating pattern and its matching points [44]. But because matches are far from perfect, the results are no better than eyeballing.)

4. Many scrolls suffered partial shrinkage over the ages, which can lead to significant errors, since the measured distance between points can be much now less than it was in the past. This explains the many overestimates. Other significant distortions of measurement can be caused by cracks in the leather or by repairs made to the scrolls.

5. Using more damage points—when available—can ameliorate anomalies caused by uneven damage.

6. In general, the differences between measurements based on the PAM or IAA images is relatively small. Despite their excellent quality, measuring damage points on the newer IAA images can yield worse results than the older PAMs, because of the continued warping of the leather in the intervening years.

7. There are difficulties in determining the scale of images, especially of the PAMs, and in particular for longer scrolls.

8. The size of the original core, if any, can only be guessed, as it clearly varies from scroll to scroll.

9. There is only an insignificant difference between the better approximation obtained using the spiral method and that of the concentric-circle method, and it is overshadowed by all the other inaccuracies. See Section S1 Appendix: Spiral Approximation in S1 File and Table 2.

10. Whether the number of turns for the circle approximation is a whole number or a simpler fractional value also makes little difference. See Section Circle Approximation and Table 2.

In short, the mathematical models commonly used to estimate the missing length of deteriorated historical scrolls have been empirically found to lead to frequent and significant errors.

We have not discussed here the reliability of Stegemann's method per se—in those cases where it is only used to reconstruct the relative placement of scattered fragments without calculating the entire length of the scroll. However, some of the problems raised here should be taken into consideration even in that usage. It is especially important to address the questions of the scaling of the images and the shrinkage of the skin. We recommend that whenever possible reconstruction methods be used in conjunction with—and as the support of—other material and textual evidence.

This article focused on the method used to reconstruct Dead Sea Scrolls, which are captured in two series of high-quality images. Fragmentary scrolls found at other locations, such as Herculaneum and Oxyrhynchus, are often reconstructed with similar methods. Many of the present outcomes should be taken in consideration by papyrologists as well. However, since each corpus has its own unique state of material preservation and history of imaging, the specific considerations will surely vary.

## Supporting information

**S1 File.**
(PDF)

## Acknowledgments

This research was made possible thanks to images provided by the Leon Levy Dead Sea Scrolls Digital Library of the Israel Antiquities Authority and by the Shrine of the Book of the Israel Museum. We thank the members of the Scripta Qumranica Electronica (SQE) project (http://qumranica.org) and the experts at the Israel Antiquities Authority and Israel Museum for their invaluable help. We thank Jonathan Ben-Dov, Bronson Brown-deVost, David Carr, Asaf Gayer, Holger Essler, and Drew Longacre for their comments on earlier drafts of this paper.

## Author Contributions

**Conceptualization:** Eshbal Ratzon, Nachum Dershowitz.

**Data curation:** Eshbal Ratzon, Nachum Dershowitz.

**Funding acquisition:** Nachum Dershowitz.

**Investigation:** Eshbal Ratzon.

**Methodology:** Eshbal Ratzon, Nachum Dershowitz.

**Software:** Nachum Dershowitz.

**Visualization:** Eshbal Ratzon, Nachum Dershowitz.

**Writing – original draft:** Eshbal Ratzon, Nachum Dershowitz.

**Writing – review & editing:** Eshbal Ratzon, Nachum Dershowitz.

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
