## [Editor Report · Decision Letter 0]

7 Aug 2020

PONE-D-20-22519

The Length of a Scroll

PLOS ONE

Dear Dr. Ratzon,

Thank you for submitting your manuscript to PLOS ONE. After careful consideration, we feel that it has merit but does not fully meet PLOS ONE’s publication criteria as it currently stands. Therefore, we invite you to submit a revised version of the manuscript that addresses the points raised during the review process.

I need the following clarification before sending your manuscript out to reviewers.

When you list three assumptions on page 4, I don’t see the most important assumption that there are no missing pieces at the end of the scroll. What if the scroll you analyze is just the first part of a longer scroll? Moreover, missing parts of the scroll are more likely happen at the end of the scroll, not in the middle.Figure 1A is not relevant to the current study: we are not interested in the width of the scroll, but its length. Therefore, the side picture that shows concentric circles must be shown.Figure 1B and Figure 1C are confusing. If the scroll can be unfolded as seen, there is no problem of measuring its length because the lateral damage does not affect the length. Are the fragments (2*pi rolls) are connected? It looks they are. But if they are, there is no problem of measuring the length of the scroll. If they are not connected the gaps must be shown. How lengths of the gaps were computed? There is some explanation on page 6 on lines 213-222 but they are very short and unclear. Are the gaps were estimated from the wedge approximation in Appendix? If so, it must be clearly stated. I believe that most errors come from how the gap’s length was estimated.Abbreviations like 11Q5 (=11QPs) are not spelled out.

I’m quite sure that you can provide clear explanations to my questions.

Looking forward to see the revised manuscript and sending it to reviewers.

We look forward to receiving your revised manuscript.

Kind regards,

Eugene Demidenko, Ph.D.

Academic Editor

PLOS ONE

---

## [Author Response · Author response to Decision Letter 0]

18 Aug 2020

We thank the editor for his comments, and propose the following clarifications:

1. "When you list three assumptions on page 4, I don’t see the most important assumption that there are no missing pieces at the end of the scroll. What if the scroll you analyze is just the first part of a longer scroll? Moreover, missing parts of the scroll are more likely happen at the end of the scroll, not in the middle."

We added a clarification at the beginning of p. 3 of the paper, namely that the Stegemann method itself only tries to estimate the length of the scroll from the preserved scroll inwards. This is an explicit assumption of the “geometric” method that we are analyzing.

2. "Figure 1A is not relevant to the current study: we are not interested in the width of the scroll, but its length. Therefore, the side picture that shows concentric circles must be show."

We moved fig. 1 (now fig. 6) near to its relevant discussion, and added an explanation of the reason for including it, which is to show the different repeating patterns of damages before and after the scroll was unrolled.

3. "Figure 1B and Figure 1C are confusing. If the scroll can be unfolded as seen, there is no problem of measuring its length because the lateral damage does not affect the length. Are the fragments (2*pi rolls) are connected? It looks they are. But if they are, there is no problem of measuring the length of the scroll. If they are not connected the gaps must be shown. How lengths of the gaps were computed? There is some explanation on page 6 on lines 213-222 but they are very short and unclear. Are the gaps were estimated from the wedge approximation in Appendix? If so, it must be clearly stated. I believe that most errors come from how the gap’s length was estimated."

We also added some clarifications for our intentions in examining this scroll. The length of extant scroll is known more or less; that was not the point. Rather, the question is whether examining part of the damage pattern can yield that length.

4. "Abbreviations like 11Q5 (=11QPs) are not spelled out."

We added full names and explanation of abbreviations on p. 4.

We would be most happy to provide any further clarifications, if needed.

Thank you very much.

Sincerely, 

Eshbal Ratzon and Nachum Dershowitz

---

## [Decision Letter · Decision Letter 1]

15 Sep 2020

The Length of a Scroll

PONE-D-20-22519R1

Dear Dr. Eshbal Ratzon,

We’re pleased to inform you that your manuscript has been judged scientifically suitable for publication and will be formally accepted for publication once it meets all outstanding technical requirements.

Kind regards,

Eugene Demidenko, Ph.D.

Academic Editor

PLOS ONE

Additional Editor Comments (optional):

Please consider the following suggestions:

1) The present title does not fully reflect the content of the paper, because you examine the damage of the roll not necessarily related to its length.  I suggest something along this line: "Quantitative evaluation of an ancient scroll damage"

2) Pay more attention to the crucial assumption that the outer part of the scroll is not missing. Although you mention this assumption in the Introduction, it deserves more attention.

3) Can't we look at how irregular the end of the scroll is in order to evaluate if the outer part is not missing?

These suggestions at at the discretion of the authors.

Reviewers' comments:

Reviewer's Responses to Questions

**Comments to the Author**

1. If the authors have adequately addressed your comments raised in a previous round of review and you feel that this manuscript is now acceptable for publication, you may indicate that here to bypass the “Comments to the Author” section, enter your conflict of interest statement in the “Confidential to Editor” section, and submit your "Accept" recommendation.

Reviewer #1: (No Response)

2. Is the manuscript technically sound, and do the data support the conclusions?

Reviewer #1: Yes

3. Has the statistical analysis been performed appropriately and rigorously? 

Reviewer #1: Yes

4. Have the authors made all data underlying the findings in their manuscript fully available?

Reviewer #1: Yes

5. Is the manuscript presented in an intelligible fashion and written in standard English?

Reviewer #1: Yes

6. Review Comments to the Author

Reviewer #1: This is a superb, highly significant article. It has quite obviously gone through multiple stages of revision, and it is convincing and professionally produced. You show an unusually comprehensive grasp of the problems surrounding images of the Dead Sea Scrolls, various methods used by classicists and Qumran specialists to estimate scroll length, and the problems with those methods. The research method chosen for the study is excellent, and the mathematical expertise applied to various models is impressive. I anticipate that this article could have a major impact on questions surrounding scroll reconstruction, particularly in study of the Dead Sea Scrolls.

In the process of reading I found just a few minor turns of phrase the might bear review. The suggestions are listed below. Otherwise, this article is quite fine, and I am glad to see it ready to appear to the broader world. The following refers to pages and lines.

1, line 9 “inherent in” not “inherent to” ??

1, line 14 “implications for” not “implications on”

1, line 23 “testimonies for” doesn’t work

2, 29 “or rather the scroll fragments”

4, line 145 “suggested” not “predicted”

7, line 237 “is known by the famous” not “is by the well-known”

9 Text after the table should read “The table”

11 middle there seems that there may be an error of some kind for the line beginning “possibility. If one chooses...”

13 issue of double damage points — still can observe pattern?

13, line 335 “a half a millimeter of possible” not “a half a millimeter possible”

13, line 338 eliminate comma before “while the scroll”

13-14 some overlap on the problem of scaling the PAM images with the earlier discussion

14, line 359 add comma before despite the better scaling”

14, line 368 “a slightly more accurate spiral approximation” is noted as if it has not been discussed previously in the article. but I believe it has.

18 “does not hold” not “need not hold”

7. PLOS authors have the option to publish the peer review history of their article (what does this mean?). If published, this will include your full peer review and any attached files.

Reviewer #1: No

---

## [Editor Report · Acceptance letter]

22 Sep 2020

PONE-D-20-22519R1 

The Length of a Scroll 

Dear Dr. Ratzon:

I'm pleased to inform you that your manuscript has been deemed suitable for publication in PLOS ONE. Congratulations! Your manuscript is now with our production department. 

Kind regards, 

on behalf of

Dr. Eugene Demidenko 

Academic Editor

PLOS ONE